# Formation of Shaped Charge Projectile in Air and Water

**DOI:** 10.3390/ma15217848

**Published:** 2022-11-07

**Authors:** Zhifan Zhang, Hailong Li, Longkan Wang, Guiyong Zhang, Zhi Zong

**Affiliations:** 1State Key Laboratory of Structural Analysis for Industrial Equipment, School of Naval Architecture Engineering, Dalian University of Technology, Dalian 116024, China; 2State Key Laboratory of Explosion Science and Technology, Beijing Institute of Technology, Beijing 100081, China; 3China Ship Research and Development Academy, Beijing 100192, China; 4Collaborative Innovation Center for Advanced Ship and Deep-Sea Exploration, Shanghai 200240, China

**Keywords:** shaped charge projectile, velocity attenuation law, underwater explosion, trans-media

## Abstract

With the improvement of the antiknock performance of warships, shaped charge warheads have been focused on and widely used to design underwater weapons. In order to cause efficient damage to warships, it is of great significance to study the formation of shaped charge projectiles in air and water. This paper uses Euler governing equations to establish numerical models of shaped charges subjected to air and underwater explosions. The formation and the movement of Explosively Formed Projectiles (EFPs) in different media for three cases: air explosion and underwater explosions with and without air cavities are discussed. First, the velocity distributions of EFPs in the formation process are discussed. Then, the empirical coefficient of the maximum head velocity of EFPs in air is obtained by simulations of air explosions of shaped charges with different types of explosives. The obtained results agree well with the practical solution, which validates the numerical model. Further, this empirical coefficient in water is deduced. After that, the evolutions of the head velocity of EFPs in different media for the above three cases are further compared and analyzed. The fitting formulas of velocity attenuation of EFPs, which form and move in different media, are gained. The obtained results can provide a theoretical basis and numerical support for the design of underwater weapons.

## 1. Introduction

With the widespread use of cabins near shipboard [1,2] and protection materials [3,4,5,6] for the design of warships, their explosion and shock resistance [7,8,9] is rapidly improved, which makes it very difficult for blast warheads to cause destructive attacks. However, due to the limitation of the dimensions of the warheads, the effect of increasing the charge weight on the improvement of the warhead power is minimal. Therefore, shaped charge warheads are gradually utilized to design the underwater weapon. In the traditional three types of shaped charges, explosively formed projectiles (EFP) [10,11] have the advantages of significant mass, small resistance, high velocity, and strong penetration ability, which are more suitable for underwater shaped charge warhead design. Therefore, it is significant to investigate the formation of EFP and its velocity attenuation in different media.

Many researchers studied the velocity attenuation law of shaped charge projectiles in air. Berner et al. [12] carried out a theoretical analysis of the flight characteristics of EFPs in air. Li et al. [13] made a theoretical analysis based on the EFP principle and flight dynamics principle and found that EFP aerodynamic resistance was significantly different when air density was different due to different temperatures. Liu et al. [14] designed a new two-wing EFP, which improved the penetration capability of the EFP. Olivera [15] proposed a numerical and analytical method for EFP maximum velocity performance estimation and verified the reliability of the analytical method through numerical simulation. Wu et al. [16] fitted the velocity attenuation equation of EFPs in air using numerical simulation. In addition, he experimentally verified the reliability of the fitting equation so that the flight distance and penetration capability of EFPs could be predicted. Du et al. [17] studied the attenuation law of the flight velocity of EFPs. However, little research about the velocity attenuation law of EFPs in water has been published.

The formation and velocity attenuation of shaped charge projectiles in water differ from those in air. Zhang et al. [18,19,20,21,22,23,24,25] systematically studied the underwater explosion and analyzed the damage of shaped charge projectiles to structures underwater. The results showed that the damage of shaped charge projectiles to structures underwater was more severe than that in the air. Cao et al. [26] studied the forming and velocity attenuation law of metallic jets in water but did not give the velocity attenuation law of metallic jets in water. According to Newton’s Second Law, Lee et al. [27] introduced velocity attenuation and resistance coefficients and presented the classical theoretical formula for fragment entry into water. Tuo et al. [28] experimentally and numerically studied the evolution of the cavity and the velocity attenuation of a high-speed projectile entering water. The above studies mainly focus on velocity attenuation of the fragment without fracture, head deformation, and mass loss. However, its shape and mass are constantly changing as it moves in water. Therefore, it is of great significance to further study the velocity attenuation law of projectiles with complex shapes and high velocity in different media, especially in the water-entry process.

In order to make the projectile keep a better shape in its formation process in water, an air cavity is utilized at the bottom of the liner. The velocity of projectile information processes in air and water entry should be investigated. Sun et al. [29,30] analyzed the change of the missile’s velocity across the medium from different incident angles and water entry speeds, but their studies focused on the low-speed interval. Wang et al. [31] analyzed the general law of EFP attenuation underwater. The effective velocity of EFP penetration in water was analyzed, but the attenuation formula was not given. Sun et al. [32] gave the optimal underwater torpedo air cavity length for underwater EFPs. However, they only considered the effect of the length of the air cavity on velocity, not whether the EFP breaks. Zhou et al. [33] studied the velocity attenuation of projectiles across the medium and established the physical model of the conical and spherical charge under the action of an underwater explosion. This model improved the residual velocity of EFPs in water but did not give the velocity attenuation equation across the medium. Mukhtar Ahmed et al. [34] recorded and calculated the velocity of the EFP by using Flash X-ray technology. The results show that the numerical simulation could reasonably predict the performance of the EFP to the underwater target. Most of the above scholars only analyzed the velocity attenuation of a projectile from air to water qualitatively, while they did not give the equation form of the velocity attenuation of projectiles.

In this paper, empirical formulas of EFPs in different media are modified or given. First, numerical models of shaped charges in air and water with/without air cavities are developed; and their formation processes are compared. After that, the effects of different charges on the maximum velocity of EFPs are discussed, with the empirical coefficient obtained in air. Based on that [35], the empirical coefficient of the maximum velocity of EFPs in water is given. Finally, the velocity attenuation law of EFPs in different media is studied. The velocity attenuation formulas of EFPs in water and from air to water are fitted by combining theoretical formulas with numerical simulation.

## 2. Basic Theory

### 2.1. Formation Velocity of Projectile in Air

An approximate analytical solution can estimate the maximum velocity of an EFP in its formation process. However, many assumptions are utilized for the analytical solution, leading to a deviation. In order to correct the analytical solution, many researchers combine experimental data with their empirical formula of the maximum velocity of EFPs in air, given by [35]:(1)$$u=1-\frac{1}{0.016\eta +0.22\sqrt{\eta }+1},$$
where $\eta =\frac{16}{27}\frac{\rho_{e}l_{p}}{\rho_{m}\xi }$; $l_{p}$ and ξ are charge thickness at the midpoint of the liner and liner thickness, respectively; $\rho_{e}$ and $\rho_{m}$ are explosive and liner densities, respectively.

It is also pointed out that the EFP velocity calculated by the above approximate analytical solution is generally 30% higher than the measured value [35]. Therefore, the actual initial velocity of an EFP in the air can be obtained by:(2)$$V_{1-A}=0.7\times D\times u,$$
where u can be solved by Equation (1) and D is the detonation velocity of the explosive.

### 2.2. Attenuation Velocity of Projectile in Water

When the shaped charge projectile with high velocity moves in water, its surface shall be covered with supercavitation. Most of its surfaces do not directly contact water, and the friction resistance is negligible. The main factor affecting the projectile’s movement is the differential pressure resistance which affects the shape of the projectile head. After moving in water for some time, its head develops into a “mushroom” shape. The head velocity of the shaped charge projectile rapidly declines. According to Newton’s Second law [36]:(3)$$m\frac{d^{2}z}{dt^{2}}=m\frac{dv}{dt}=-\frac{1}{2}\rho_{w}A_{0}C_{d}V_{t}{}^{2},$$
where m is the mass of the shaped charge projectile; z is the distance that the shaped charge projectile advances; $V_{t}$ is the projectile velocity at any time; $A_{0}$ is the projected area of the head of the shaped charge projectile in contact with water, and $C_{d}$ is the resistance coefficient related to the cavitation number. Although the cavitation number and the resistance coefficient changes with the movement of the projectile, they are so small that they are set to a constant in this paper.

By integrating Equation (3), we get [36]:(4)$$V_{t}=\frac{V_{0}}{1+V_{0}\beta t},$$
where $V_{0}$ is the initial velocity when it enters into the water, and constant β is the velocity attenuation coefficient and is defined as [36]:(5)$$\beta =\rho_{w}AC_{d}/2m,$$
where β is related to the density of seawater; $\rho_{w}$ the head area of the projectile; A the resistance coefficient, $C_{d}$ and the mass m of the projectile. It is defined as a constant in this paper.

### 2.3. Attenuation Velocity of Projectile from Air to Water

Assume that a projectile with density $\rho_{m}$ moves with velocity $u_{x}$ in a Newtonian fluid with a viscosity coefficient of μ. The resistance on the projectile is F. It is assumed that the surface of the projectile is smooth and rotates asymmetrically, and its gravity, cavitation resistance, and temperature are not considered. According to the momentum equation and Newton’s Second Law, the force and the mass can be given by [37]:(6)$$Fdt=mu_{x},$$
(7)$$F=-Mdu_{x}/dt,$$
(8)$$M=\rho_{m}SL,m=\rho Sdx,$$
where M is the mass of the projectile, and m is the liquid mass acting on the front of the projectile head in time dt. Yang et al. [37] derived the velocity attenuation equation through the above formula and fitted it with polynomials. However, the physical meaning of the independent variables in the formula is ambiguous. On this basis, taking time as the independent variable and fitting with polynomials, this approach obtains good verification, and then the fitting formula of projectile velocity with time is expressed as [37]:(9)$$u_{t}=u_{0}\exp\left(-At-Bt^{2}+Ct^{3}\right),$$
where $u_{t}$ is the projectile velocity at any time; $u_{0}$ is the initial velocity when it goes from air to water; t is the time, and A, B, and C are resistance constants.

### 2.4. Numerical Theory

#### 2.4.1. Fluid Governing Equation

The Euler algorithm is used to simulate projectile formation in different media in this paper. The Euler grid is fixed, with materials transported in Figure 1. The primary calculation process is divided into three steps in AUTODYN. Conservation equations of mass, momentum, and energy are given by [38]:(10)$$\frac{\partial \rho }{\partial t}+\frac{\partial \rho u}{\partial x}+\frac{\partial \rho v}{\partial y}=0,$$
(11)$$\begin{array}{l} \frac{\partial \rho v}{\partial t}+\frac{\partial \rho uv}{\partial x}+\frac{\partial \left(\rho v^{2}+P\right)}{\partial y}=0\\ \frac{\partial \rho u}{\partial t}+\frac{\partial \left(\rho u^{2}+P\right)}{\partial x}+\frac{\partial \rho uv}{\partial y}=0 \end{array}$$
(12)$$\frac{\partial E}{\partial t}+\frac{\partial u(E+P)}{\partial x}+\frac{\partial v(E+P)}{\partial y}=0,$$
where x and y are coordinates, and ρ,v,u,E, and P are the density radial velocity, axial velocity, internal energy, and pressure of the fluid, respectively.

#### 2.4.2. Equation of State

(1)Equation of state for water

The shock equation of state is adopted for water, expressed as [38]:(13)$$U_{S}=C_{0}+S_{1}u+S_{2}u^{2},$$
where $U_{s}$ is the shock wave velocity: u is the particle velocity; $C_{0}$, $S_{1}$ and $S_{2}$ are constants, and the specific values are set as $C_{0}=1647\left(m/s\right)$, $S_{1}=1.921,S_{2}=0$.

(2)Equation of state for air

The ideal gas equation is adopted for air, given by [38]:(14)$$p_{air\;}=(\gamma -1)\rho_{air\;}e_{air\;},$$
where $p_{air}$ is air pressure; $\rho_{air}$ is air density, adiabatic constant γ=1.4, specific energy $e_{air}=2.068\times 10^{5}J/\mathrm{Kg}$.

(3)Equation of state for metal liner

Copper is used as the material of metal lining. The linear equation is used for the equation of state, and the Johnson-Cook equation is used for the strength model, expressed as [38]:(15)$$Y=\left[A+B\varepsilon_{p}^{n}\right]\left[1+C\ln\varepsilon_{p}^{*}\right]\left[1-T_{H}^{m}\right],$$
where Y is equivalent stress; A is initial yield stress; B is the hardening constant; $\varepsilon_{p}^{*}$ is the plastic strain rate; n is the hardening index; C is the strain rate constant; m is the thermal softening index, and $T_{H}$ is dimensionless temperature. The detailed parameters of the Johnson-Cook equation for copper are listed in Table 1.

(4)Equation of state for explosives

The JWL equation is adopted for explosives, given by [38]:(16)$$p_{e}=A\left(1-\frac{\omega }{R_{1}\bar{V}}\right)e^{-R_{1}\bar{V}}+B\left(1-\frac{\omega }{R_{2}\bar{V}}\right)e^{-R_{2}\bar{V}}+\frac{\omega E}{\bar{V}},$$
where $\bar{V}=\rho_{b0}/\rho_{b}$, $\rho_{b}$, and $\rho_{b0}$ are the density of detonation products and their initial density; E is the internal energy of explosive per unit volume; A, B, $R_{1}$, $R_{2}$, and w are constants which are obtained by a specific experiment, and $p_{e}$ is explosive detonation pressure. The detailed parameters of the JWL equation with different types of explosives are listed in Table 2 [38].

## 3. Formation Process of Shaped Charge Projectiles in Different Media

### 3.1. Numerical Model

In order to study the formation law of shaped charge projectiles in different media, two-dimensional axisymmetric models of air and underwater explosions of shaped charges with spherical-segment liners were established. Denote three cases—air explosion and underwater explosions with and without air cavity—as Cases 1, 2, and 3, respectively. Four types of explosives were chosen: TNT, comp B, C4, and HMX. A numerical model of the air explosion of the shaped charge for Case 1 is shown in Figure 2. The charge had a height *L* of 40 mm and a diameter *D* of 20 mm. The liner was made of copper with variable thickness. Its inner and outer diameters were *r* = 13.99 mm and *R* = 12.20 mm. The dimension of the air cavity was variational. In order to avoid the reflection of shockwaves after reaching the boundary, the flow-out boundary was applied as a fluid boundary. The sub-option and preferred material for the flow-out boundary condition were flow-out (Euler) and all equal, respectively. The mesh size was determined after a convergence analysis.

The numerical model of the underwater explosion of the shaped charge is similar to that of the air explosion, as shown in Figure 3.

A light torpedo has an air cavity in its actual design. Therefore, a numerical simulation model of the underwater explosion of a shaped charge with an air cavity is developed in Figure 4. The length of this air cavity *d* dramatically affected the formation and velocity of the projectile. Sun et al. [32] found that when its length was three times the charge radius, the shape of the EFP in the formation process was better. In order to find out a better length of air cavity in this paper, three cases with *d* from two to four times the charge radius were chosen in Table 3.

### 3.2. Convergence Analysis

To ensure the reliability of Euler’s algorithm, the velocity and morphology of the shaped charge projectile were simulated in this section. The experimental [39] and simulated values both agree well, as shown in Table 4 and Figure 5.

In order to obtain a reasonable mesh size, a convergence analysis was carried out. Head X-velocities of shaped charge projectiles with different grid sizes and numbers were illustrated in Table 5 and Figure 6, respectively. The obtained results show that the head velocity with a grid size of 0.2 mm×0.2 mm was similar to those of 0.1 mm×0.1 mm and 0.12 mm×0.12 mm. Taking calculation accuracy and efficiency fully into consideration, the grid size of 0.2 mm×0.2 mm is used for the simulation in this paper.

### 3.3. Formation Process of Shaped Charge Projectile

Then, the formation processes of shaped charge projectiles in air and water with and without air cavities were further analyzed.

#### 3.3.1. Case 1: Air Explosion

Firstly, the formation process of a shaped charge projectile in the air was analyzed. The velocity distribution of the projectile at different times is shown in Figure 7. At *t* = 5 μs, the detonation wave arrived at the top of the liner, with a plastic deformation caused. At *t* = 10 μs, with the shockwave effect, the liner was completely crushed, with an EFP initially formed. At *t* = 15 μs, an EFP was fully formed, and its head velocity peaked at approximately 1700 m/s. Due to the velocity gradient from the front to the back of the EFP, it was stretched, and its head and pestle could be distinguished. The EFP could fly smoothly in the air if its gravity and air resistance were ignored.

#### 3.3.2. Case 2: Water Explosion without Air Cavity

Then, the formation process of the shaped charge projectile in water was analyzed. The velocity distribution at different times is shown in Figure 8. At *t =* 5 μs, the detonation wave reached the liner top, with plastic deformation. At *t =* 10 μs, the liner began to turn over. Due to the great resistance effect of water, the shape of the EFP developed into a “crescent moon,” which is different from that of the air in Figure 7. As the EFP moved in the water, its head shape kept stable at *t =* 25 μs. However, with the movement of the EFP in water, its head was worn, which led to mass loss and velocity decrease. Penetration performance decreased as a result.

#### 3.3.3. Cases 3–5: Water Explosion with Air Cavity

Finally, the formation process of the shaped charge projectile, which moves from air to water, was analyzed. Three cases with lengths d of the air cavity of twice, three, and four times larger of charge radius are discussed in this section, namely Cases 3–5, respectively. Numerical results for velocity distributions of these three cases are illustrated in Figure 8, Figure 9 and Figure 10, respectively.

The velocity distribution for Case 3 Is shown In Figure 9. At *t =* 15 μs, it can be seen that the EFP was not completely formed while it moved from air to water. Due to the water resistance, the head of EFP was worn. At *t =* 30 μs, the EFP began to break. Due to the large velocity gradient between the front and the rear of the EFP, it was overstretched, with multiple fractures formed. More fractures were found at *t =* 40 μs and 50 μs, which decreased the penetration performance of EFP.

The velocity distribution for Case 4 is shown in Figure 10. At *t* = 15 μs, a short EFP was formed stably. At *t =* 20 μs, the head of EFP entered the water and began to be worn, with a cavity generated around it in the fluid. Mass loss of the EFP is also found, and the head of the EFP is flattened. At *t* = 30 μs, the shape of the head of the EFP developed into a “mushroom.” At *t* = 40 μs and 50 μs, the velocity gradient of the EFP was small so that fewer fractures were formed than that of Case 3.

The velocity distribution for Case 5 is shown in Figure 11. Although the EFP had been formed before it entered water, the velocity gradient of the EFP was more significant than that of Case 4, which also caused more fractures at *t* = 40 μs and 50 μs. In conclusion, when the length of the air cavity is three times of charge radius, a shaped charge projectile with better velocity and shape can be formed.

#### 3.3.4. Results Analysis and Discussion

After analyzing the formation processes of EFPs in different media, the comparison results show that the medium has an excellent effect on the formation of EFPs. A short and thick EFP is formed in air. The shaped charge projectile is turned over and develops into a “crescent moon” shape in the water. As for the case of a water explosion with an air cavity, the initial shape of an EFP before it arrives in water is similar to that in air. However, its shape gradually becomes a “mushroom” after its head arrives at the water. Due to the velocity gradient, the EFP breaks into many fractures. In addition, the effect of the length of the air cavity on the formation of EFPs is discussed. It can be found that when the length is three times the charge radius, such variables as tensile length, fracture, and water-entry velocity of the EFP are better than those of the other two cases.

## 4. Maximum Head Velocity of Projectile in Air and Water

### 4.1. Coefficient Modification of Head Velocity of Projectile in Air

According to the empirical formula in Section 2.1, the empirical coefficient of head velocity is set to 0.7 when the projectile forms in the air. Based on the air explosion model of a shaped charge in Section 3.3.1, the maximum head velocities of projectiles with four types of charge materials are discussed in this section. The empirical coefficients are numerically obtained in Table 6. Taking the average empirical coefficient of 0.647, the modified formula can be obtained as follows
(17)$$V_{1-A}=0.647\times D\times u,$$

Then, evolutions of velocity with different types of charge materials are further analyzed in Figure 12. The projectile is formed in the microsecond time scale, with its head velocity up to the peak value. After that, due to air resistance, the velocity slightly decreases. This decrease is affected by many factors, such as the windward area of the projectile, liner density, air density, etc. The detailed attenuation law of EFP flight in air was analyzed by Du et al. [17], which shall be given in Section 5.1 in detail.

### 4.2. Reduction Coefficient of Head Velocity of Projectile in Water

On the basis of the empirical formula of the maximum head velocity of the projectile in air, the formula in water is numerically deduced in this section. Evolutions of head velocity EFPs in water with different types of charges are shown in Figure 13. It can be seen that although their maximum velocities are so different, they share a similar attenuation law. Firstly, their velocity sharply decreases and then slowly declines. In this section, the maximum velocity of EFPs formed in water is studied, with an empirical coefficient for estimating the maximum velocity of EFPs given.

After validating the empirical coefficient of the head velocity of EFPs in air, similar numerical models of shaped charges subjected to underwater explosions with different types of charges are established. The obtained maximum velocities of EFPs and empirical coefficients are listed in Table 7. It is found that the empirical coefficient ranges from 0.455 to 0.476. Taking the average empirical constant of 0.462, the empirical formula of the initial head velocity of EFPs in water is obtained by:(18)$$V_{1-W}=0.462\times D\times u,$$

## 5. Effects of Media on the Evolution of Velocity

### 5.1. Velocity Attenuation Law of Projectiles in Air

It is found that such factors as flight distance, shape, the density of projectiles, etc., affect the residual velocity of projectiles [8]. Because the flight velocity of the projectile is much larger than the speed of sound, its weight is relatively small, and the air resistance is far greater than its weight, and the influence of gravity on the speed of EFP is ignored in the calculation, so the flight trajectory of the EFP can be regarded as a straight line, and its motion equation is [17]:(19)$$q_{f}=\frac{dV}{dt}=-C_{D}\frac{\rho_{0}}{2}A_{s}H(Y)V^{2},$$
where $q_{f}$ is the actual weight of the projectile; $C_{D}$ is the air resistance coefficient; $A_{S}$ is the windward area of the projectile; H(Y) is the relative air density at height *Y*; $\rho_{0}$ is the ground air density, and V is the instantaneous flight speed of the projectile.

Among them, the resistance coefficient varies with the shape and flight velocity of the projectile. In order to obtain the analytical expression of residual velocity with distance, linear standardization is usually used for the solution of *C_D_*, which is based on the measured results. At the second stage in air, the velocity attenuation formula of the projectile is given by [17]:(20)$$V_{2-A}=V_{1-A}\exp\left[-\frac{C_{D}H(Y)\rho A_{s}}{2q_{f}}r\right],$$
where $V_{1-A}$ is the initial velocity in the first stage, r is the radius of the projectile, and ρ is the density of the air at the location.

### 5.2. Velocity Attenuation Law of Projectile in Water

HMX is found to work best in air and water. Therefore, it is used as an explosive in the following sections. Then, the velocity attenuation law of a projectile in water without an air cavity is analyzed. The formation and propagation process of a projectile can be divided into three stages: acceleration, rapid decay, and slow decay stages, as shown in Figure 14. In the first stage, the velocity increases linearly and peaks at 1300 m/s at about 0.1 μs. After that, the velocity sharply decreases in the second stage, with an attenuation coefficient. In the third stage, it slowly declines. Based on this, the velocity attenuation law of a projectile under different charges is further analyzed. According to Figure 13, it can be preliminarily judged that the underwater velocity attenuation is similar and has specific laws. It is found that the four cases with different types of charge share a similar evolution of velocity in Section 4.2. Next, the detailed velocity attenuation law in the second and third stages is analyzed.

#### 5.2.1. Velocity Attenuation Law in the Second Stage

The velocity attenuation law in the second stage is first analyzed in Table 8. With the increase of explosive detonation velocity, both the maximum head velocity of the projectile and the attenuation coefficient increase. This indicates that the greater the initial velocity of the underwater projectile is, the greater its instantaneous attenuation velocity also is, which leads to a larger attenuation coefficient. Numerical results show that the attenuation coefficient ranges from 0.569 to 0.630, with an average attenuation percentage of about 58.8%. Therefore, the velocity attenuation formula in the second stage can be obtained by:(21)$$V_{2-W}=0.588\times V_{1-W},$$

#### 5.2.2. Velocity Attenuation Law in the Third Stage

The velocity attenuation law in the third stage is analyzed in this section. Take (0.0128 ms, 801 m/s) as the starting point of the third stage in Figure 15. The numerical results for the evolution of velocity in the third stage are fitted according to Equation (4) in Section 2.2, given by:(22)$$V_{3-W}=\frac{V_{2-W}}{1+0.01486V_{2-W}t},V_{2-W}=800.947\;m/s$$
The theoretical formula fits well with the numerical simulation, but the numerical results fluctuate. The velocity attenuation coefficient β is set as a constant when fitting according to the theoretical formula. However, the velocity attenuation coefficient β varies in the actual process because of the fluctuation of numerical results. When a shaped charge projectile moves in water, the mass falls off. The head gradually became a “crescent moon” shape, and the projectile’s head-on area changed. In the process of penetration, both of them changed simultaneously. According to Equation (5), the velocity attenuation coefficient β also changes. This paper selected three points, A, B, and C, with large fluctuations to further analyze the specific reasons, as shown in Figure 15 and Table 9.

There are some differences between theoretical fitting and numerical simulation in Figure 15. At the beginning of detonation, the projectile’s mass does not change during 0~0.1 ms in Figure 16. However, the projectile head area is the main factor affecting the attenuation coefficient β. Detonation waves force the head area of the projectile to decrease during the extreme time of detonation of the explosive. Subsequently, the area increases due to the influence of water resistance. That is, the attenuation coefficient β decreases first and then increases. However, in this paper, the attenuation coefficient β is taken as a constant, which results in the theoretical velocity being small at first and then prominent in the range of 0~0.1 ms. The head area of the projectile is stable, and the head gradually becomes a “crescent moon” after *t =* 0.1 ms, as shown in Figure 8. At this time, the projectile mass is the main influencing factor of the attenuation coefficient β. The increase in attenuation coefficient β is caused by the shedding of projectile mass. However, this paper takes the attenuation coefficient β as a constant, which results in the theoretical velocity being less than the numerical simulation velocity. In this paper, three points with relatively large fluctuations are marked as A, B, and C, respectively, and the recorded data are shown in Table 9. First, the velocity fluctuation range of theoretical fitting is 20–30 m/s. Secondly, it has basically lost its penetration ability [31] when projectile velocity drops to approximately 200. Therefore, a 20–30 m/s velocity error does not affect the evaluation of the damage degree. The fitting formula of the third stage is reliable.

### 5.3. Velocity Attenuation Law of Projectile from Air to Water

In order to obtain a better shape and velocity of the projectile, it is more suitable to add an air cavity inside the liner for the design of a lightweight torpedo rather than a formation in water. The water entry of the projectile should be investigated in this process. Therefore, the velocity attenuation law during the water entry process is further analyzed in this section. According to the results in Section 3.3.3, the better length of the air cavity is about three times larger than the charge radius. Thus, the evolution of the projectile’s velocity for Case 3 is illustrated in Figure 17.

Three stages are included: acceleration, pitting, and water entry stages. Due to the fracture and collision of the projectile, the water entry stage is further subdivided into fracture and collision fluctuation stages, respectively. The projectile forms in the air in the first stage, and its velocity increases linearly. In the second pit stage in the BCD region in Figure 17, the projectile velocity first decreases and then climbs slightly. As for the third stage-water entry stage, the velocity decay is slow and fluctuates due to fracture and collision of the shaped charge projectile.

#### 5.3.1. Velocity Analysis in Pit Stage

The pitting stage is a unique phenomenon of the projectile, which forms in water. Four points, A, B, C, and D, are marked in Figure 17, and their specific values of velocities are shown in Table 10. Meanwhile, pressure distributions when the projectile arrives at the above four points are shown in Figure 18. The wave load should be of concern because it is the main energy that overwhelms the liner at the moment of the burst. At *t* = 5 μs, a detonation wave is generated and propagates in water. Besides, with the effect of a detonation wave, the velocity of the liner peaks in a very short time. At *t* = 10 μs, the shockwave propagates from the water to the air cavity and begins to dissipate., so the head velocity of the projectile decreases slightly. However, At *t* = 15 μs, with the continuous effect on the projectile, its velocity increases slightly. At *t* = 20 μs, the projectile begins to enter the water. After that, the shockwave has little effect on the velocity of the projectile. The pit stage is basically over. Then, due to different media after the air cavity, the attenuation law is different. If the projectile moves in the air, it shall fly stably, with the velocity decreasing slightly, as in Figure 18. However, if it moves from air to water, the velocity rapidly declines. The effect of media on the velocity of the projectile in the pit stage is discussed in the next section. Finally, it is worth mentioning that the medium of wave load propagation in pure air and water (cases 1 and 2) remains unchanged, so the phenomenon of the pit stage does not occur.

Three cases are listed and discussed in Table 11. Evolutions of velocity for Cases 4 and 6 are first compared. Both of them showed a slight decline and climb, resulting in a concave phenomenon. Under the two working conditions, the time and speed are basically the same in Figure 19. This result indicates that if the length of the air cavity is sufficient to shape the projectile, then the water entry velocity of the projectile is essentially the same. Even if the length of the air cavity is increased further, the velocity of the projectile will not increase. Besides, evolutions of velocity for Cases 1 and 6 are compared. The maximum velocity of the projectile in water with an infinite air cavity is slightly higher than that only in air, as shown in Figure 19. The reason is that the shockwave dissipates quickly in the air, while the shockwave propagates faster in water, and the effect is more substantial than that in the air.

#### 5.3.2. Velocity Analysis in Water Entry Stage

At the water entry stage, (0.019 ms, 1746.775 m/s) is taken as the initial point. After the time reaches zero, the velocity of the projectile in the water entry stage is fitted according to Equation (9) in Section 2.3, with A=3.880,B=127.286,C=652.968. Therefore, the evolution of velocity can be obtained by:(23)$$u_{t}=1746.775\exp\left(-3.880t-127.286t^{2}+652.968t^{3}\right)$$

Fluctuations are found in the numerical and fitting curves in the water entry stage. Three fluctuation points, D, E, and F, are chosen in Figure 20, with the fracture fluctuation stage of $t_{D}=0.041\;\mathrm{ms}$ and $t_{E}=0.071\;\mathrm{ms}$ collision fluctuation stages of $t_{F}=0.085\;\mathrm{ms}$. It can be seen that the fracture begins to be caused at point D, and the curve fluctuates accordingly. The projectile during the phase between D and E breaks, with its head worn, and its head shape gradually develops into a “mushroom.” However, its tail does not directly contact the water after the air cavity, with a higher velocity than the head. The tail catches up with the head at point F and begins to impact the head, with the velocity slightly increasing. After that, with the merge of the head and tail, the velocity gradually stays stable and drops to about 400 m/s. As a result, the projectile basically does not have penetration capability [31].

In order to further verify the reliability of the theoretical formula, three points, G, H, and I, in Figure 20, with large fluctuations, are selected for error analysis in Table 12. The maximum velocity fluctuation is 54 m/s, and the maximum error percentage is approximately 8%, validating the theoretical formula. After that, the shape of the projectile stays stable without fractures forming anymore, which corresponds to the velocity attenuation law in water. Finally, the residual velocity of the projectile decreases to approximately 400 m/s, and the projectile has basically lost its penetration ability.

## 6. Conclusions

Based on the theoretical formula of the head velocity of the shaped charge projectile in the formation process, the Euler method was used to establish the air and underwater explosion models of a shaped charge with and without air cavity, with shapes of the projectile analyzed in different media. The empirical coefficient of head velocity attenuation in the formation process in water is given. The variation law of the head velocity of projectile in different media is discussed. The specific conclusions are given as follows:A shaped charge projectile formed in air is short, thick, and dense while it turns over to be a “crescent moon” in water and develops into a “mushroom” shape from the air cavity to water. Due to the velocity gradient, fractures are found when the projectile enters and moves in the water. When the length of the air cavity is lower or larger than three times of charge radius, the projectile cannot be completely formed or easily fractured. Therefore, it is suggested to make the length of the air cavity three times larger than the charge radius;Velocity attenuation laws of shaped charge projectiles with four types of explosives in air and water are discussed. Results show that the empirical coefficients of maximum velocity in air and water are 0.647 and 0.462, respectively. The head velocity of a projectile in water can be divided into three stages: acceleration, rapid decay, and slow decay. The higher the maximum head velocity of a projectile is, the greater the percentage of velocity attenuation is in the rapid decay stage. The residual velocity is about 60% of the maximum head velocity. The theoretical fitting formula is given in the slow decay stage, and its results agree well with the numerical ones. The maximum error of head velocity is only about 30 m/s, which proves the high reliability of the theoretical fitting formula;The shaped charge projectile forms in the air cavity and then enters the water. Its head velocity includes acceleration, pit, and water entry stages. Because of the fracture and collision of the projectile, the water-entry stage is divided into fracture and collision stages. The pitting stage is a unique phenomenon of a projectile in water. Its velocity tendency shows that the velocity first declines and then increases and eventually stays steady. The theoretical fitting formula of the head velocity of a projectile in the water-entry stage is given. The maximum error between the theoretical and numerical results for a projectile’s head velocity is lower than 8.1%, which validates the theoretical fitting formula. Besides, the fluctuations are found in the numerical results caused by the fracture and the projectile collision.

## Figures and Tables

**Figure 1 materials-15-07848-f001:**
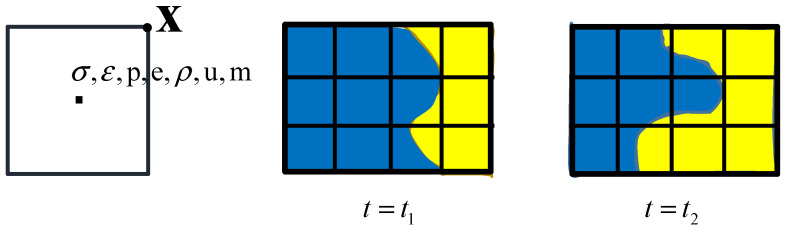
Materials transported in Euler.

**Figure 2 materials-15-07848-f002:**
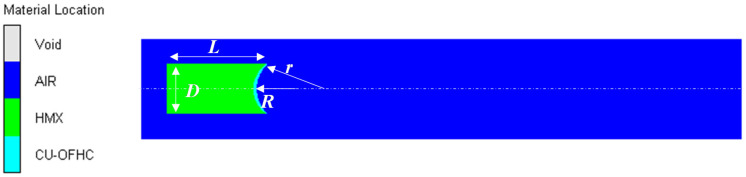
Numerical model of the air explosion of the shaped charge.

**Figure 3 materials-15-07848-f003:**
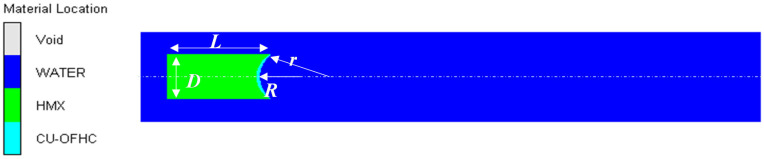
Numerical model of the underwater explosion of a shaped charge without an air cavity.

**Figure 4 materials-15-07848-f004:**
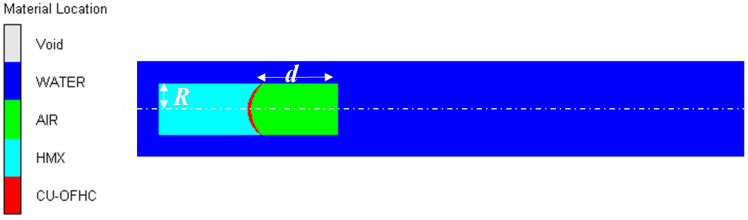
Numerical model of the underwater explosion of a shaped charge with air cavity.

**Figure 5 materials-15-07848-f005:**
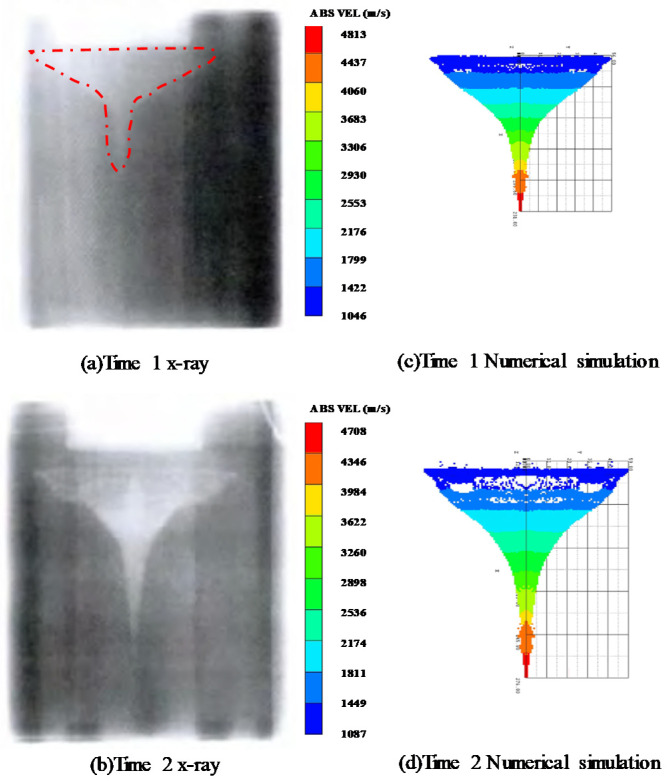
Comparison of experimental results [39] and numerical simulation.

**Figure 6 materials-15-07848-f006:**
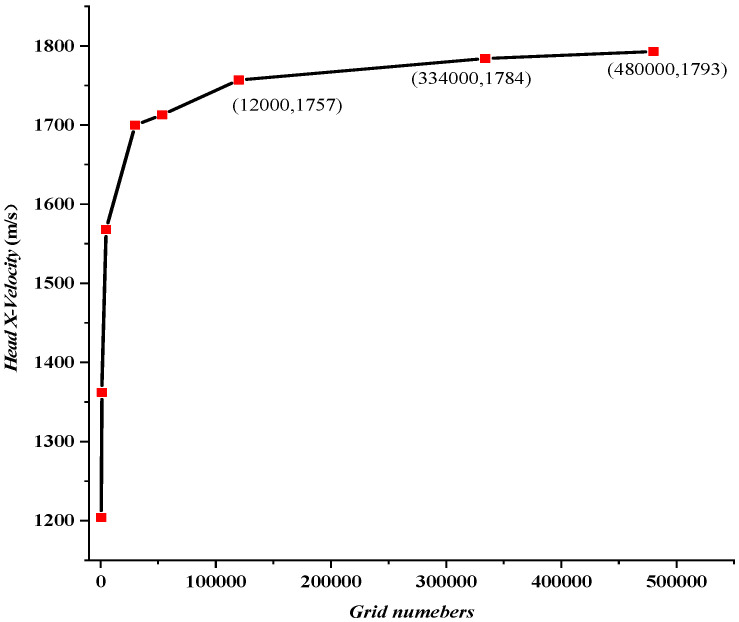
Head X-velocity of a projectile with different grid numbers.

**Figure 7 materials-15-07848-f007:**
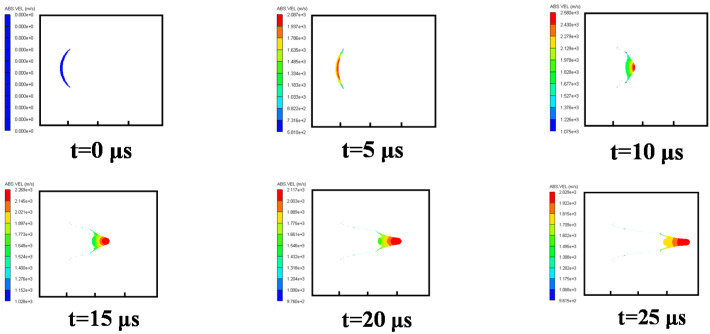
Velocity distribution of EFPs in a formation process in air.

**Figure 8 materials-15-07848-f008:**
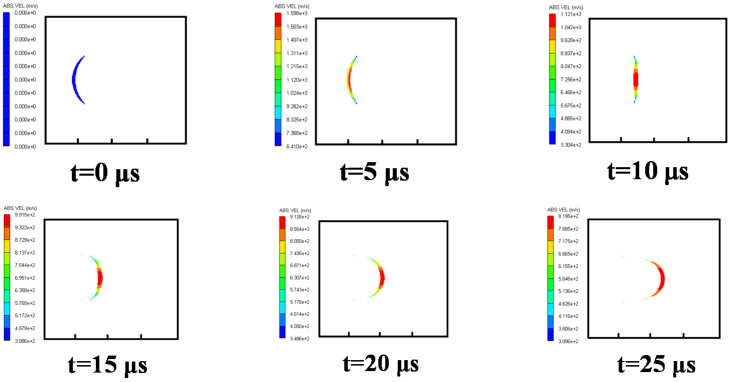
Velocity distribution of the EFP in a formation process in water.

**Figure 9 materials-15-07848-f009:**
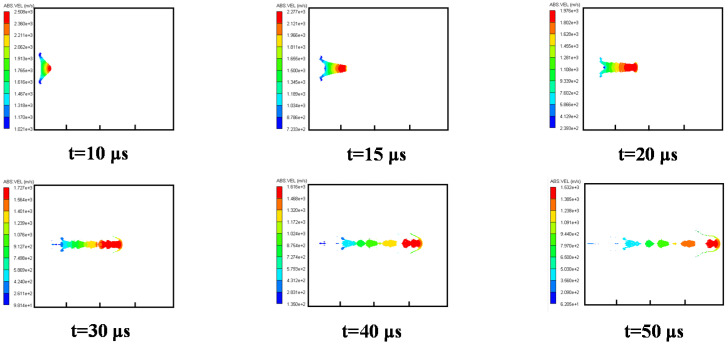
Velocity distribution of the EFP for Case 3.

**Figure 10 materials-15-07848-f010:**
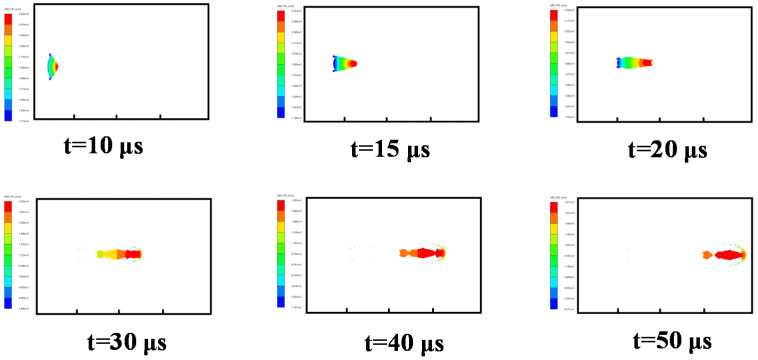
Velocity distribution of the EFP for Case 4.

**Figure 11 materials-15-07848-f011:**
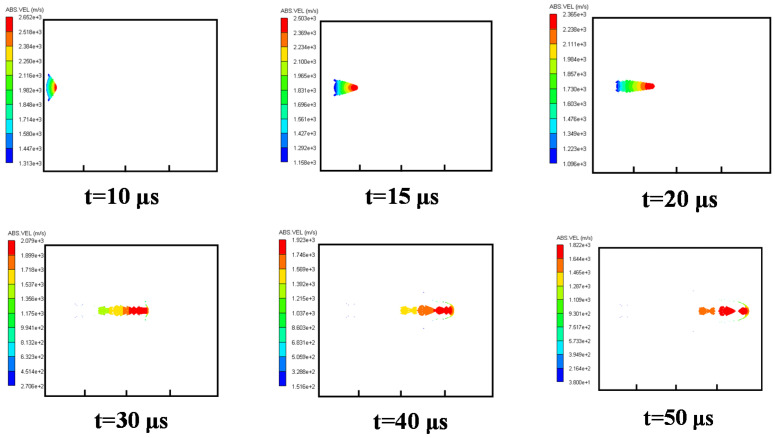
Velocity distribution of the EFP for Case 5.

**Figure 12 materials-15-07848-f012:**
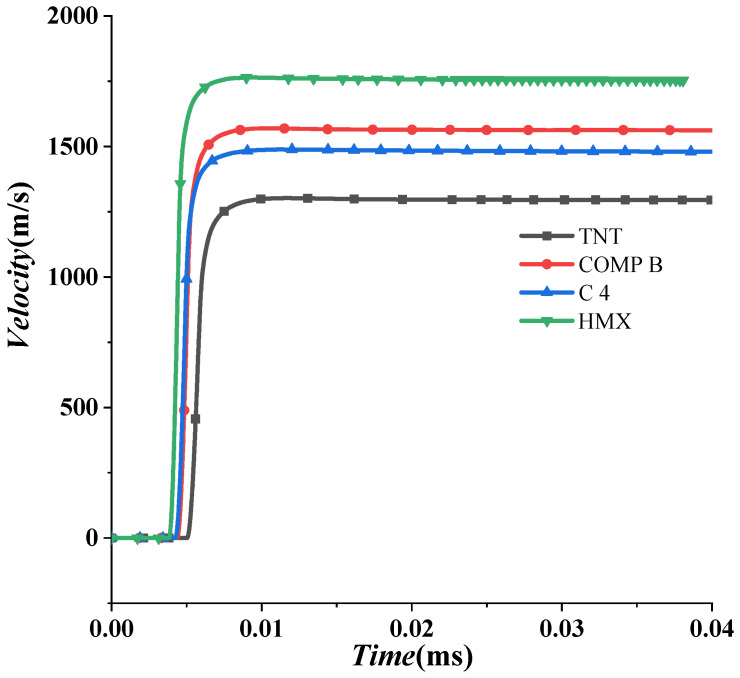
Evolutions of the maximum head velocity of projectiles with different types of charges in air.

**Figure 13 materials-15-07848-f013:**
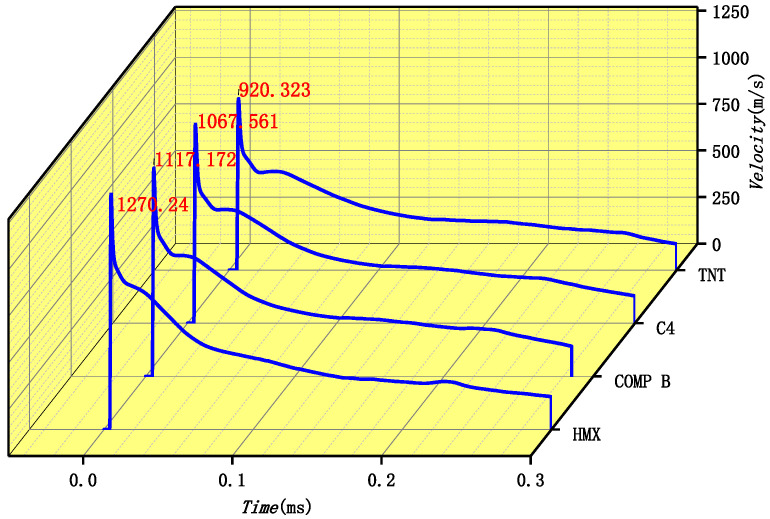
Evolution of the velocity of EFPs with different types of charge.

**Figure 14 materials-15-07848-f014:**
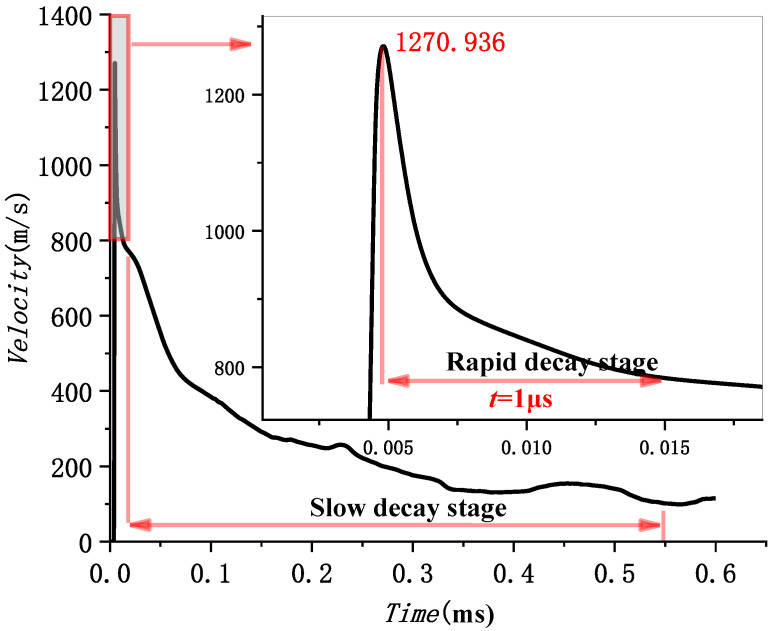
Evolution of velocity of projectile in water without air cavity.

**Figure 15 materials-15-07848-f015:**
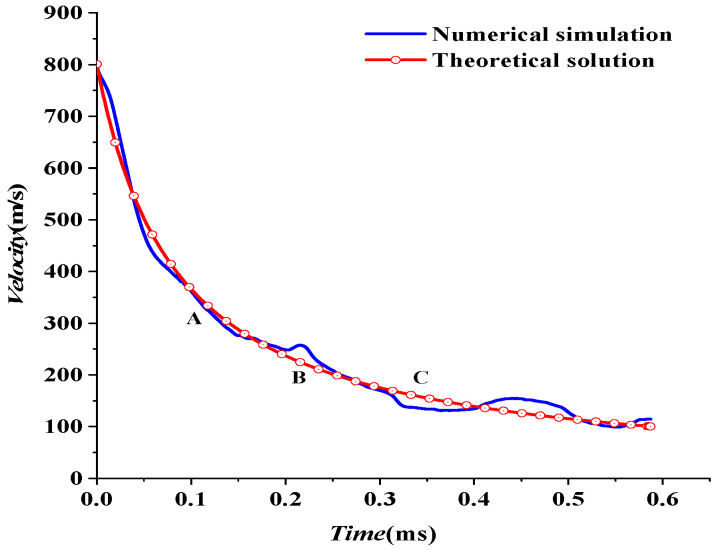
The fitting curve of underwater velocity attenuation of the EFP at the third stage.

**Figure 16 materials-15-07848-f016:**
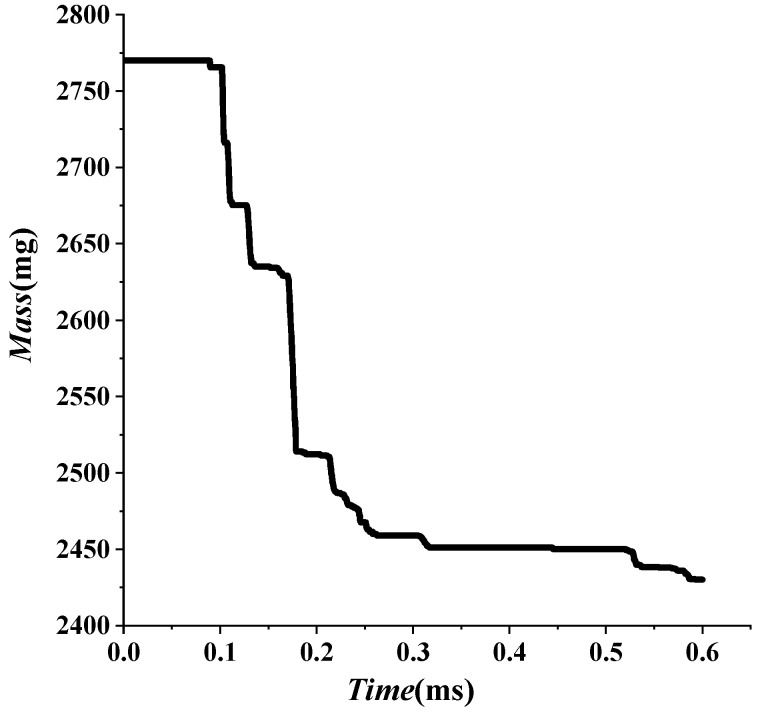
Evolution of mass of the shaped charge projectile in water.

**Figure 17 materials-15-07848-f017:**
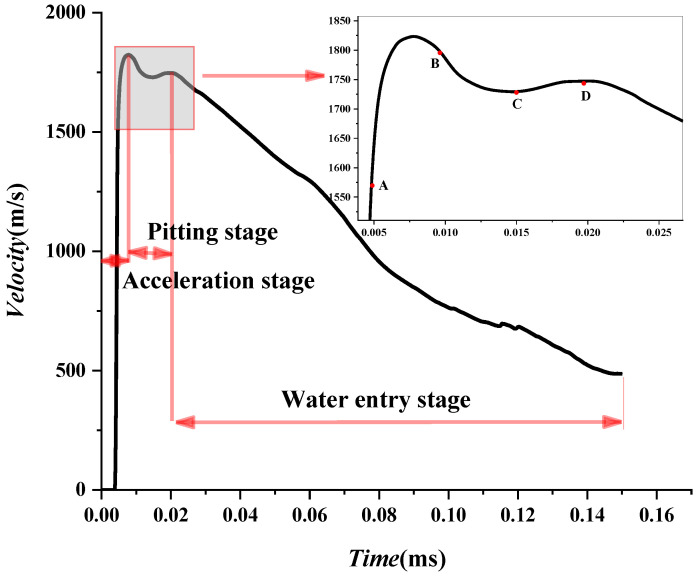
Evolution of the projectile entering water from air.

**Figure 18 materials-15-07848-f018:**
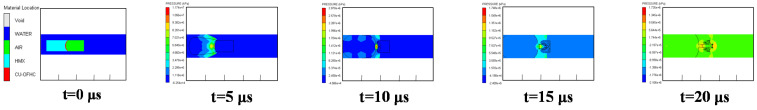
Shock wave propagation in the pit stage.

**Figure 19 materials-15-07848-f019:**
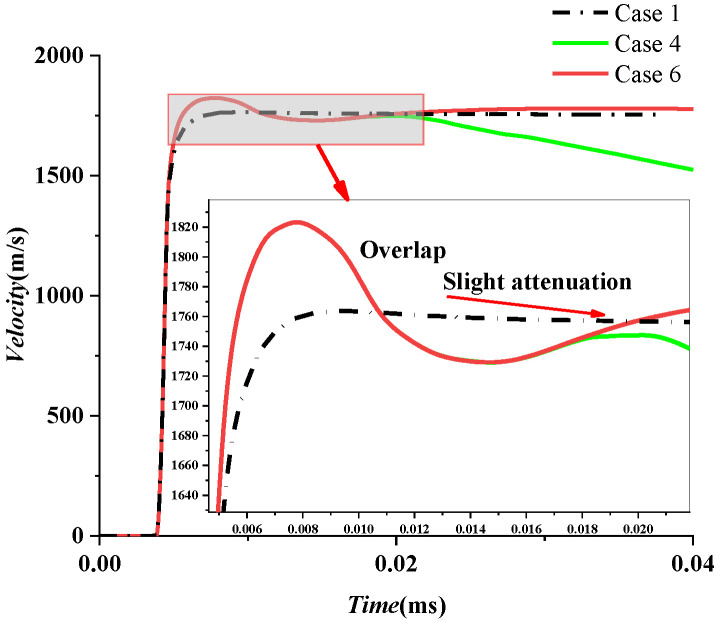
Comparison analysis diagram of the pitting stage and maximum velocity.

**Figure 20 materials-15-07848-f020:**
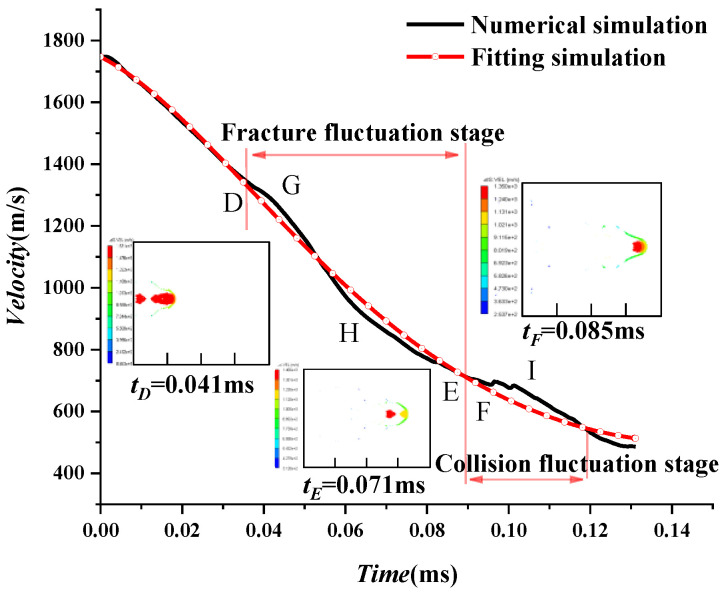
Attenuation curve of the shaped charge projectile entering water.

**Table 1 materials-15-07848-t001:** Parameters of the constitutive model of the liner [38].

*A* MPa	*B* MPa	*N*	*C*	*M*	*Bulk* *Modulus*/KPa	*Reference* *Temperature*/K	*Specific Heat* (J·kg^−1^K^−1^)
90	292	0.31	0.025	1.09	$1.29\times 10^{8}$	300	383

**Table 2 materials-15-07848-t002:** Parameters of the JWL equation of different types of explosives [38].

Type	*A* GPa	*B* GPa	$R_{1}$	$R_{2}$	ω	$\begin{array}{c} \rho \\ \mathrm{kg}\cdot m^{-3} \end{array}$	$\begin{array}{c} D_{CJ}\\ m\cdot s^{-1} \end{array}$	$\begin{array}{c} E\\ \mathrm{GJ}\cdot m^{-3} \end{array}$	$\begin{array}{c} p_{CJ}\\ \mathrm{GPa} \end{array}$
TNT	373.77	3.75	4.15	0.90	0.35	1630	6930	6.0	21.0
COMPB	524.23	7.68	4.20	1.10	0.34	1717	7980	8.5	29.5
C4	609.77	12.95	4.50	1.40	0.25	1601	8193	9.00	28.0
HMX	778.28	7.07	4.20	1.00	0.30	1891	9110	10.5	42.0

**Table 3 materials-15-07848-t003:** Cases for different numerical models.

Cases	Media	Air Cavity Length
1	Air	-
2	Water without air cavity	-
3	Water with air cavity	Twice the charge radius
4	Water with air cavity	Three times the charge radius
5	Water with air cavity	Four times the charge radius

**Table 4 materials-15-07848-t004:** Comparison of parameters between experimental [39] and simulated values.

Results	Time1 (μs)	Time2 (μs)	Head Velocity (km/s)	Tail Velocity (km/s)	Length (mm)
Experiment [39]	41.25	50.65	5.23	1.13	134.20
Simulation	40.00	50.00	4.71	1.09	145.00
Error/%	−3.00%	−1.28%	−9.94%	−3.54%	8.05%

**Table 5 materials-15-07848-t005:** Simulated EFP head velocity under different grid sizes.

Grid Size (mm)	Grid Numbers	Head X-Velocity (m/s)
3.00 × 3.00	648	1204
2.00 × 2.00	1200	1362
1.00 × 1.00	4800	1568
0.40 × 0.40	30,000	1700
0.30 × 0.30	53,600	1713
0.20 × 0.20	120,000	1757
0.12 × 0.12	334,000	1784
0.10 × 0.10	480,000	1793

**Table 6 materials-15-07848-t006:** Empirical coefficient of the head velocity of projectiles with different types of charge materials in air.

Material	*D* (m/s)	$\rho_{e}\left(g/{\mathrm{cm}}^{3}\right)$	$\rho_{m}\left(g/{\mathrm{cm}}^{3}\right)$	Simulation Velocity (m/s)	Empirical Coefficient	Average Empirical Coefficient
TNT	6930	1.630	8.960	1302.58	0.653	0.647
COMP B	7980	1.717	8.960	1569.55	0.669	
C4	8193	1.601	8.960	1488.81	0.635	
HMX	9110	1.891	8.960	1763.94	0.634	

**Table 7 materials-15-07848-t007:** Empirical coefficient of the head velocity of projectiles with different types of charge materials in water.

Material	D(m/s)	$\rho_{e}\left(g/{\mathrm{cm}}^{3}\right)$	$\rho_{m}\left(g/{\mathrm{cm}}^{3}\right)$	Simulation Velocity (m/s)	Empirical Coefficient	Average Empirical Coefficient
TNT	6930	1.630	8.960	920.428	0.461	0.462
C4	8193	1.601	8.960	1067.892	0.456	
COMP B	7980	1.717	8.960	1118.419	0.477	
HMX	9110	1.891	8.960	1270.936	0.457	

**Table 8 materials-15-07848-t008:** Velocity attenuation in the second stage.

Material	Simulation Maximum Velocity (m/s)	Sharply Decrease to Velocity (m/s)	Attenuation Factor	Average Attenuation Factor
TNT	920.428	524.312	0.570	0.588
C4	1067.892	610.582	0.572	
COMP B	1118.419	649.375	0.581	
HMX	1270.936	800.947	0.630	

**Table 9 materials-15-07848-t009:** Error analysis of head velocity.

	Point A	Point B	Point C
Time (ms)	0.218	0.350	0.450
Numerical simulation (m/s)	256.456	134.028	153.957
Theoretical equation (m/s)	222.448	155.046	126.015
Velocity error	−34.008	21.018	−27.941
Percentage error	−13.260%	15.682%	−18.149%

**Table 10 materials-15-07848-t010:** Parameters of data points in the pit stage.

	Point A	Point B	Point C	Point D
Time (μs)	5	10	15	20
Velocity (m/s)	1638.864	1786.006	1729.389	1747.401

**Table 11 materials-15-07848-t011:** Cases discussed in the pit stage.

Cases	1	4	6
Media	air	Water with air cavity	Water with infinite air cavity
Lengths of air cavity	-	Three times the charge radius	Infinite

**Table 12 materials-15-07848-t012:** Velocity error analysis.

	Point G	Point H	Point I
Time (ms)	0.041	0.070	0.102
Numerical simulation (m/s)	1292.232	858.142	682.358
Theoretical equation (m/s)	1253.264	892.540	627.438
Velocity error	−38.968	34.398	−54.920
Percentage error	−3.02%	4.01%	−8.05%

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
