# Peer review of "Formation of Shaped Charge Projectile in Air and Water"

_materials, 2022, doi:10.3390/ma15217848_

Round 1

Reviewer 1 Report

The manuscript presents a numerical models of shaped charges subjected to air and underwater explosions. Velocity attenuation of EFP in different media is analysed and fitting equations are derived, which can be useful in the design of weapons. Methodology applied in the research is appropriate and discussion and conclusions are supported by numerical modeling and experimental data. The manuscript can be useful for those interested in design of EFP weapons.

A small remark is that the manuscript is bit too lengthy, but I recommend publishing in its present form.

I'm not competent in English language, but it seems to me that proofreading is needed

Reviewer 2 Report

1. The methods and software used in this study need to be clarified.

2. Also, the boundary conditions used in the simulation need to be clarified more detail. Example: 

- The effect of pressure of the fluid respectively (P) is given in the numerical theory. But when the results came out, no effect was found. 

- And the ambient temperature effect caused by friction (by great speed) is not mentioned. If omitted, are the simulation results consistent with the experiment?

Reviewer 3 Report

 The authors of the paper entitled “Formation of shaped charge projectile in air and water” modified or gave empirical formulas of explosively formed projectiles (EFP) in different media.  They also fitted the velocity attenuation equations of EFP in water and from air to water. The studies performed are relevant to the research of interest and would bring significant benefit to the understanding of the formation and velocity attenuation of shaped charge projectiles in water. However, there are some rooms to improve this paper.

 Eq (1). The definition of the parameters expressed in this equation is not given.

It is not clear where these parameters “l and x are charge thickness at the midpoint of the liner and liner thickness respectively; re and rm are explosive and liner densities, respectively” take place ( page 2, lines 94-95.

The authors did not present Eq. to evaluate the detonation velocity D.

The lack of information on the numerical determination of empirical coefficients.

V is the instantaneous flight speed of the projectile and the maximum head velocity of EFP, i.i.e two different parameters are marked similarly. It is confusing.

Velocity attenuation is marked as Vr and V2.

There is no definition of Vmax  ( Eq.21, Line 345).

The conclusion must be more specific. For example, what is meant by “better in the sentence “It is found that when the length of the air cavity is three times of the charging radius such results as the shape, non-fracture, and water entry velocity of the projectile are better. (Line 464)

English needs attention. For examples:

 It is written, “this paper obtains good verification” (line 135). But a paper itself could not obtain good verification.

Page 3, line 133, “the” must be instead “The”.

 It is not clear whether the lengths of the air cavity is larger or smaller than the charge radius. See line 239, sentence “Three cases with lengths d of the air cavity of twice, three, and four times of charge radius are discussed in this section, namely Cases 3 -5, respectively”.

The empirical coefficients could not numerically obtain in Table ( Line 276)., etc.

Round 2

Reviewer 2 Report

Dear authors

The specific modifications and explanations are okay,

Thank you